# Case Studies in Polar Bear (*Ursus maritimus*) Sperm Collection and Cryopreservation Techniques

**DOI:** 10.3390/ani12040430

**Published:** 2022-02-11

**Authors:** Jessye Wojtusik, Terri L. Roth, Erin Curry

**Affiliations:** Cincinnati Zoo and Botanical Garden, Center for Conservation and Research of Endangered Wildlife, 3400 Vine St., Cincinnati, OH 45220, USA; jessye.wojtusik@omahazoo.com (J.W.); terri.roth@cincinnatizoo.org (T.L.R.)

**Keywords:** electroejaculation, endangered species, gamete rescue, polar bear, sperm cryopreservation, sperm rescue, testicular tumor, urethral catheterization

## Abstract

**Simple Summary:**

Polar bears are threatened by habitat loss, decreased food availability, and reduced reproductive success due to climate change. Zoo populations can support species survival through preservation of genetic diversity and maintenance of insurance populations, but in the US, the zoo polar bear population is currently not sustainable. The development of sperm collection and cryopreservation can help to support the population by providing the biomaterial needed for assisted reproductive techniques, such as artificial insemination. However, these procedures are not well described for polar bears. Data from 38 opportunistic sperm collections, that were conducted between 2011 and 2021, were assessed to establish best practices to date for collecting and preserving polar bear sperm. The information gathered demonstrates that urethral catheterization is an efficient method of sperm collection, sperm can be rescued postmortem from the vasa deferentia and epididymides, and polar bear sperm collection appears to be most effective during the breeding season. Furthermore, polar bear sperm can survive cryopreservation. Further studies will optimize these techniques, but this summary provides information that is immediately applicable to enhancing sample collection and cryopreservation success that could support the long-term genetic management of polar bears in zoos.

**Abstract:**

Assisted reproductive technologies can aid conservation efforts via support of *ex situ* population management and preservation of genetic material. Data from 38 sperm collection attempts from 17 polar bears (1–5 procedures/bear) were evaluated. Sample collections were attempted via electroejaculation (EEJ; *n* = 6), urethral catheterization (UC; *n* = 25), or sperm rescue (SR; *n* = 7) during the breeding season (Jan. 1-May 21; *n* = 27) and nonbreeding season (May 22-Dec. 31; *n* = 11). Sperm retrieval was successful in 1 EEJ (16.7%), 18 UC (72.0%) and 4 SR (57.1%) collections. Initial sperm motility and viability were 50.0% and 77.0% for EEJ, 64.3 ± 7.4% and 80.9 ± 3.8% for UC, and 56.7 ± 8.8% and 80.5 ± 0.5% for SR. UC and SR were more likely to be successful during the breeding season (84.2–100%) than the nonbreeding season (25.0–33.3%). Testicular tumors were observed in four males (57%) during SR. In total, 13 samples were cryopreserved (*n* = 1 EEJ, 9 UC, and 3 SR) with egg-yolk-based equine extender (EQ) or OptiXcell (OP). For both extenders, post-thaw motility and viability were reduced by 20–60% and 30–65%, respectively. Further efforts to optimize procedures are warranted, but this summary provides data useful for enhancing the success of polar bear sperm collection and cryopreservation.

## 1. Introduction

Polar bears (*Ursus maritimus*) were first classified as threatened by extinction by the International Union for Conservation of Nature (IUCN) in 1965 [1]. Despite global conservation efforts, this classification remains, as an increase in reproductive failure and cub mortality and a simultaneous decrease in habitat and food availability threaten the survival of multiple subpopulations [1,2,3,4]. *Ex situ* populations support species survival via preservation of genetic diversity and by promoting conservation education and public awareness of species status and the global impact of climate change on biodiversity [5,6]. However, the *ex situ* population of polar bears in the US is not sustainable and in decline, birth rates do not compensate for mortality rates [7], and federal regulations limit supplementing *ex situ* populations with wild imports.

Sperm collection and cryopreservation may be useful to evaluate fertility in breeding males, preserve genetic material for use in future assisted reproductive efforts, and reduce risks associated with moving animals between institutions [8]. These techniques are established in some species of bears, including brown bears [9], black bears [10,11,12], and giant pandas [13,14]. However, the literature describing such procedures in polar bears is limited. Testes from hunted polar bears were examined though sperm characteristics were not reported [15,16]. Curry et al. [17] briefly described the use of urethral catheterization (UC) and electroejaculation (EEJ) in polar bears for use in conjunction with artificial insemination; however, the sample size was limited, and further investigation is needed to optimize techniques.

Over the last decade, our lab has conducted 38 opportunistic sperm collection attempts on 17 male polar bears within the North American *ex situ* population. EEJ and UC collections were conducted to assess and monitor the fertility status of male polar bears in this population and were performed opportunistically in conjunction with veterinary procedures. Sperm rescues (SRs) were conducted following natural death or euthanasia due to terminal or age-related diseases. Though data collection did not occur within the confines of a defined and replicated study, the data are useful in providing insights into which techniques may be effective for the collection and cryopreservation of polar bear sperm and provide groundwork crucial for the development of optimized protocols for these techniques. Data are presented as a series of case studies describing EEJ, UC, SR, and sperm cryopreservation. Our goals were to coalesce outcomes from these polar bear sperm collections and to (1) evaluate the efficacy of each collection technique (EEJ, UC, and SR), (2) assess if the likelihood of a successful collection was impacted by timing (breeding or nonbreeding season), (3) define sperm characteristics, and 4) conduct preliminary sperm cryopreservation testing.

## 2. Materials and Methods

Over the last decade (2011–2021), 38 procedures were conducted to collect sperm from 17 polar bears (1-5 procedures/bear; Table 1) managed in 17 North American zoological institutions. The males ranged in age from 3 to 32 years old (mean ± SEM: 16.2 ± 1.4 y) and weighed 286 to 544 kg. Collections occurred both during the breeding season (*n* = 27; 1 January–21 May) and the nonbreeding season (*n* = 11; 22 May–31 December) [18]. Samples were collected from live bears via EEJ (*n* = 6) [17] or UC (*n* = 25) [17,19] and postmortem via SR (*n* = 7) [12,20]. All protocols were reviewed and approved by the Cincinnati Zoo and Botanical Garden’s Animal Care and Use Committee (IACUC). Each collection occurred opportunistically in conjunction with scheduled veterinary procedures. Anesthetic drug choice was at the discretion of the veterinary staff at each participating institution. Combinations of butorphanol, carfentanil, diazepam, isoflurane, ketamine, midazolam, medetomidine, dexmedetomidine, and/or telazol were used. A collection was defined as successful if sperm were present in the sample regardless of sample quality.

Sample quality was assessed immediately following retrieval. Sample motility (% motile) and progressive status (scale of 0–5; 5 = motile sperm displayed rapid linear progression) were evaluated at 200× magnification with phase optics on prewarmed slides (37 °C). Sperm concentration was determined using a hematocytometer (American Optical, Buffalo, NY 14215, USA). A morphological assessment was conducted at 400× magnification with phase optics. Sperm were classified as normal if absent of any primary (including cephalic abnormalities or damaged midpieces) or secondary defects (including bent midpieces or tails and proximal or distal droplets). Viability was assessed using eosin–nigrosin live–dead exclusion stain (Jorgensen Laboratories, Inc., Loveland, CA 80538, USA). For the UC samples, viability data were only collected for 7 procedures and morphological assessments were only conducted for 11 procedures. For the SR samples, viability was not recorded for one sample.

### 2.1. Electroejaculation (EEJ)

EEJ was conducted a total of six times with five bears (1–2 procedures/individual). Testes were palpated and length and width were determined by holding the testes flush with the skin and using calipers to take measurements prior to collection. A rectal probe, 3.3 cm in diameter with three electrodes (7.5 cm × 0.5 cm) and an electroejaculator (P-T Electronics, Boring, OR 97009, USA) were used to administer stimuli (60–165) of 2–10 V over the course of 2–5 series [17]. Probe size was selected based on previous reports in polar bears [17] with consideration of similar species [21], and size and anatomy of the rectum to avoid forcible placement or damage to the rectal membrane. Amperage ranged from 25 mAmps to 350 mAmps. There was a five-minute period of rest between each series. As no ejaculation occurred, semen was washed from the urethral opening into sterile plastic sample cups and stored until evaluation in an insulated container with a warmed (37 °C) gel pack, to protect the sperm from cold shock.

### 2.2. Urethral Catheterization (UC)

UC was conducted a total of 25 times with 14 bears (1–5 procedures/individual) [17]. Testes were palpated and measured as described for EEJ. The penis was extruded from the prepuce by pushing the caudal baculum cranially. A sterile, lubricated polypropylene suction catheter (8 Fr; Covidien, Mansfield, MA 02048, USA) or PVC nasogastric tube (8 Fr; Neomed, Woodstock, GA 30188, USA) was inserted ~40 cm into the urethra and left in place for approximately one minute. A syringe (3 mL) was attached to the free end, and the catheter was slowly retracted while maintaining slight (~0.5 mL) negative pressure. The lumen of the catheter was flushed with an extender (~1 mL) into a 15 mL tube, and samples were stored in an insulated container with a warmed (37 °C) gel pack until evaluation. The catheterization was repeated one to two additional times, each with a new sterile catheter/tube.

### 2.3. Sperm Rescue (SR)

Sperm rescue from the vasa deferentia and epididymides was attempted opportunistically in seven polar bears ranging in age from 11 to 32 years old. Upon the natural death or humane euthanasia of the males, testes, epididymides, and vasa deferentia were removed from the body cavity by veterinary staff at the housing institution. For six of the seven SR attempts, the collected tissues were wrapped in gauze soaked in saline and shipped overnight with frozen icepacks to the authors. Care was made to ensure tissue did not come into direct contact with the ice packs by wrapping them in towels. Upon arrival, testis length and width were recorded. Testis weight was also recorded for four of the seven collections. Vasa deferentia were dissected away from the surrounding tissue and flushed with an extender by inserting a sterile 23 g needle into the end adjacent to the testis [12,20]. The sample was collected into a sterile culture dish and the flush was repeated 2–3 times using approximately 250–500 µL of extender per flush. The surface of the epididymal tails was gently sliced open with multiple shallow cuts using a scalpel blade and then rinsed with an extender (~500 µL) into a separate culture dish [12,20]. The testes were sliced, and slides were pressed on the dissected surface to visualize any spermatozoa that may be within the seminiferous tubules. Sections of each testis, epididymis, and vas deferens from each male subject to SR were collected and fixed in 10% formalin. The tissues were submitted to Michigan State University Veterinary Diagnostic Laboratory (Lansing, MI 4809, USA) for histological assessment, and six of the seven were assessed for inclusion in this manuscript.

### 2.4. Sperm Cryopreservation

In total, 13 samples (*n* = 1 EEJ from 1 individual, 9 UC from 5 individuals, and 3 SR from 3 individuals) were processed for cryopreservation. Of those successful UC collections not cryopreserved, six were of poor motility (≤30%) or low concentration, and three were contaminated with urine. One of the successful SR collections was used immediately for artificial insemination instead of being processed for cryopreservation. A portion of the sperm from 12 of the 13 samples was extended in equine semen extender (EQ; containing lactose (5.5% *v/v*), disodium EDTA (0.25% *w/v*), egg yolk (20% *v/v*), glucose (1.5% *w/v*), Equex STM (0.25% *v/v*; Nova Chemical, Moon Township, PA 15108, USA), 25 iu penicillin G mL^−1^, 25 iu streptomycin mL^−1^) [22] and cryopreserved using conventional freezing techniques previously described for other species [23]. The sample was diluted with EQ (~300 × 10^6^ sperm/mL), cooled in a water bath to 4 °C (~1 h), and then, the diluted sample was diluted further in a 1:1 stepwise manner (25, 25, 50% *v/v* every 20 min), with chilled (4 °C) EQ containing 8% glycerol. The samples were then incubated at 4 °C for 1 h, loaded into 0.25 cc straws, lowered into a charged dry shipper (42 cm; 3.6 L capacity; Chart MVE Biomedical, Ball Ground, GA 30107, USA) for 10 min, and then plunged into liquid nitrogen.

In preliminary trials, one SR sample and a portion of one of the UC samples were processed with the extender, OptiXcell (OP; containing carbohydrate, mineral salts, buffer, antioxidants, glycerol (12.8%), phospholipids, water, antibiotics: gentamicin, tylosin, lincomycin, specomycin). OptiXcell was prepared according to the manufacturer’s instructions (IMV Technologies USA, Maple Grove, MN 55369, USA) and used as previously described [24]. The UC sample was diluted 1:1 with OP for a final glycerol concentration of ~6%. The SR sample was collected directly into OP, and the final glycerol concentration was estimated to be ~10–12%. The samples were loaded into 0.25 cc straws, chilled for 1 h at 4 °C, placed in a charged dry shipper for 10 min, and then plunged into liquid nitrogen. All samples were thawed for 10 sec in room temperature (RT) air and then submerged for 20 s in a water bath (37 °C), for post-thaw assessment.

### 2.5. Statistical Analysis

A Fisher’s exact test was used to evaluate the success rate between EEJ and UC and between seasons (breeding vs. nonbreeding) within the UC and SR groups separately. As SR is performed postmortem, the success rate was not compared with collections from live bears. The limited number of EEJ collections conducted during the nonbreeding season (*n* = 1) prevented analysis within this group. Sperm characteristic values from multiple collections of the same individual were averaged prior to inclusion in the population average. Data from multiple collections from the same individual and testis length, width, and weight were averaged between left and right sides before the average size was calculated. As testes were measured after removal from the body cavity for SR procedures, these measurements were reported separately from the EEJ and UC data. Measurements from testes found to contain tumors were not included in the calculations. Data are reported as mean ± standard error of the mean (SEM), and significance is defined as *p* < 0.05. Statistical analyses were conducted using SPSS for Windows (Version 27; IBM Corporation, Armonk, NY 10504, USA).

## 3. Results

Sperm collection methods per individual bear are outlined in Table 1. UC was successful in 18 out of 25 collections (72.0%), EEJ was successful in 1 out of 6 collections (16.7%), and SR was successful in 4 out of 7 collections (57.1%). UC was more likely to be successful (result in sperm collection) than EEJ (*p* = 0.022). The one successful collection via EEJ was conducted during the breeding season, but the impact of the season within the EEJ group could not be statistically assessed, as only one procedure was conducted during the nonbreeding season. Within the UC group, the collection was more likely (*p* = 0.032) to be successful if conducted during the breeding season (16 out of 19; 84.2%) than the nonbreeding season (2 out of 6; 33.3%). The three unsuccessful UC collections during the breeding season were conducted between January 21 and February 26. The two successful collections conducted in the nonbreeding season were collected from the same individual (in June and October) who had a testicular tumor. When data from two males (3 collections total) confirmed at necropsy to have testicular tumors were removed from the analysis, the impact of season on the likelihood of success increased (*p* = 0.005). Testis length in bears collected via EEJ and UC ranged 45–103 mm (mean: 79.9 ± 2.4 mm) and width ranged 31–61 mm (mean: 47.5 ± 1.8 mm).

For the seven bears collected via SR, testis length ranged 55–82 mm (mean: 69.4 ± 3.3), width ranged 36–54 mm (mean: 45.5 ± 1.8), and weight ranged 42.5–69.8 g (mean: 55.5 ± 4.7 g); testes found to contain tumors were not included in calculations of averages. All three SR conducted during the breeding season were successful (100%), and one collection out of four (25%) conducted in the nonbreeding season was successful (Table 1); season was not found to significantly impact the likelihood of success (*p* > 0.05). The one successful SR during the nonbreeding season was conducted in June. Three males appeared to be aspermic, as no sperm cells were observed in any of the flushes collected from the vasa deferentia or epididymides or in smears of dissected testicular tissue from these bears. All three aspermic males were collected during the nonbreeding season. Histological examination revealed indicators of seasonal atrophy including limited spermatogenesis and increased connective tissue in two of the three aspermic males; the third aspermic male was not assessed. Testicular tumors were found in the testes of four (57%) of the males (aged 24–32 y/o) including two of the aspermic males (*n* = 5 testis; 1–2 tumors/bear). Testicular size appeared normal for two of the males with tumors, but in the other two, the afflicted testes were ~1.5–2.5x the size of the nonafflicted and up to 6x the weight (Figure 1). Histology of the two spermic males without tumors appeared normal.

Sperm characteristics of samples collected via all three methods are reported in Table 2. Three of the UC collections considered successful due to the presence of sperm were contaminated with urine and consequently lost motility and viability soon after collection. Data from the contaminated collections were not included in calculations of general sperm characteristics. Prefreeze and post-thaw sperm motility, progression, and viability are in Table 3. Morphology within all groups did not differ more than ± 5% between initial and post-thaw time points.

## 4. Discussion

This manuscript contains the most comprehensive compilation of polar bear semen collection data published to date and the first report of successful sperm cryopreservation and postmortem sperm rescue in polar bears. The data described herein support the immediate application of these technologies for preserving the extant genetic diversity of this species while concurrently providing the groundwork for optimizing cryopreservation techniques and extender choices.

Sperm motility, progression, and viability were comparable to values reported in other ursids [13,19,25]. However, this cohort of polar bears appear to have less morphologically normal sperm than some other bear species [13,19]. Sperm collected via UC from black bears demonstrated fewer morphological abnormalities than sperm collected via EEJ [26], so the difference between polar bears and other bear species may not be associated with the collection technique. There were more morphologically normal sperm in the EEJ and UC samples than those collected via SR, which is not surprising, as sperm rescued from the epididymides have greater numbers of cytoplasmic droplets [11].

### 4.1. Electroejaculation vs. Urethral Catheterization

Similar to a report in black bears [26], UC proved more successful than EEJ for sperm collection in polar bears. It should be noted that the successful EEJ described in this study did not result in a noticeable ejaculation. Instead, a small volume of sperm was washed from the urethral opening with an extender at the end of the procedure. Therefore, EEJ appears more successful in bear species other than polar bears, resulting in sperm production in up to 50% of collection attempts [19,26] versus ~17% reported herein. Polar bears are most closely related to brown bears but have several adaptations supporting their survival in arctic conditions, including adipose deposits surrounding organs [27]. Adipose is nonconductive and has a high electrical impedance, and it is, therefore, possible that EEJ is less likely to be successful in polar bears. The technique may have impacted outcomes as well; however, all practitioners were well versed in the use of these methods in a variety of species. Probe size and upper voltage were selected conservatively, to minimize the risk of trauma to the bears. In other bear species, better sperm quality parameters were observed in samples collected via UC versus EEJ [26]. EEJ sample size was not adequate for comparing sample quality between the two collection methods in this study.

UC is generally preferred by practitioners, as it is less invasive than EEJ, quicker to perform, and does not require the expensive and specific equipment needed for EEJ [28,29]. The successful use of UC may be dependent upon the use of medetomidine, an alpha-2-adrenergic agonist, within the anesthesia protocol. Alpha-adrenoreceptor agonists have been shown to stimulate contractions of the vasa deferentia in rats [30], which, in turn, may facilitate the movement of semen into the urethra. Only one male (#16) in this summary was anesthetized without the use of medetomidine, and therefore, it is difficult to conclude whether medetomidine use is strictly necessary when conducting UC for polar bears, but it is commonly used when conducting UC in other species [28,29].

### 4.2. Impact of Season

Polar bears are seasonal breeders, and similar to black bears and brown bears, males produce greater concentrations of testosterone during the breeding season and experience seasonal changes in spermatogenesis [15,16,31,32]. This seasonality was reflected in the greater success rate of collections during the breeding season versus during the nonbreeding season. Additionally, testicular tissue from two males collected via SR during the nonbreeding season displayed indicators of seasonal atrophy, including limited spermatogenesis and increased connective tissue. It is possible that the observed changes in tissue composition are age related instead of seasonal. However, wild polar bears have been shown to enter a period of inactive spermatogenesis during the nonbreeding season [15,16], and the histological findings of this summary support the idea that spermatogenesis is diminished during this time.

It is not surprising that UC (*n* = 1) and SR (*n* = 1) were successful when conducted a few weeks after the defined breeding season, as fecal hormone monitoring has shown females display estrus as late as June [33], although these are outliers in the *ex situ* population. Additionally, in some locations, wild bears are known to breed well into the early summer months [34,35]. Another out-of-season UC collection occurred in October. Testosterone production generally returns to baseline in most male bears by this time of year [18], and females are entering dens; therefore, no breeding behavior should be occurring, which implies that there is no physiological reason for males to be producing sperm at this time. This male passed away five years following the successful October collection, and upon necropsy, it was revealed that there was a tumor within one of the testes. Testicular tumors have also been reported in both geriatric black bears [36] and brown bears [37]. Some types of testicular tumors in humans are associated with an increase in testosterone production [38]. Therefore, it is possible this individual’s testicular tumor was promoting testosterone production and, therefore, stimulating sperm production out of season; however, it is unknown whether the tumor was present at the time of the collection.

The process of spermatogenesis, which can span weeks, relies on adequate testosterone production [39,40]. Testosterone concentrations in male polar bears begin to rise early in the year (~first week of January) [18], which is when wild polar bears in Greenland demonstrate a shift from a period of inactive spermatogenesis, spanning from July to January, to a period of active spermatogenesis (February to June) [16]. Therefore, it is possible that sperm could not be collected early in the breeding season if spermatogenesis was still diminished for those individuals. There are anecdotal reports of females conceiving as early as January, so some male bears must have sufficient sperm in midwinter, but individual variation exists in the timing of spermatogenesis. Regardless, based on our findings, sperm collection procedures are recommended between March and May, when possible.

### 4.3. Sperm Cryopreservation

The loss of 40–50% of sperm motility regardless of extender used is similar to results reported in other studies of bear semen cryopreservation [9,41] that have led to research on alternative freezing methods and extenders to increase cryosurvival and post-thaw quality parameters for ursid species [21,42]. In this study, the greatest number of samples was processed using EQ. The use of semen extenders containing animal products, such as egg yolk or skim milk, is common practice for many mammalian species, including bears [25,42,43], though there have been some investigations into the use of animal protein-free options for cryopreserving brown bear semen [44,45]. OP may have potential as an alternative to EQ, as the one UC sample cryopreserved with OP performed similarly to EQ samples. However, the SR sample cryopreserved with OP experienced a complete loss of motility post-thaw. The final glycerol concentration for the SR sample was 10–12%, which may have been cytotoxic. SR samples are collected directly into the extender, so care should be made to use minimal extender volume or to find an appropriate diluent that can be used for the initial collection to avoid such a high glycerol content. These data are preliminary, and additional evaluation of OP for use in polar bear sperm cryopreservation is needed.

### 4.4. Testicular Tumors

Testicular tumors were found in four bears, three of which were the oldest bears in the study (29–32 y/o), suggesting tumor development may be age related, consistent with reports of geriatric black bears and brown bears [35,36]. Unfortunately, the timing of the onset of tumor development cannot be determined. Testicular size appeared normal for one of the males with tumors, but in the other two, the afflicted testes were ~1.5–2.5x the size of the nonafflicted and up to 6x the weight. One of the three males was still producing sperm, and a 28 y/o bear was successfully collected via UC; therefore, it appears that tumor development and age do not necessarily impact spermatogenesis. However, histological assessment of the tissue did not reveal tumor origin, which may have been in the Sertoli or interstitial cells, and without an exact diagnosis, it is difficult to accurately assess the physiological impact on the health or reproductive potential of each afflicted bear. Tissue degradation made assessment difficult, and the use of a preservative other than formalin is recommended for future efforts.

### 4.5. Limitations

These data were collected opportunistically over the span of a decade, with the primary goals of evaluating fertility in an *ex situ* population of polar bears and banking samples for use in assisted reproductive procedures. Inherent limitations of this summary should be considered when interpreting results. As the data were compiled retrospectively, the design does not conform to the traditional scientific method, sample sizes are limited, and assessments were conducted by multiple individuals over many years, and therefore, strict consistency was not achieved. For example, viability was not assessed for all samples. These limitations associated with the retrospective nature of this study are compounded by the threatened status of the focus species—polar bears. The *ex situ* polar bear population is limited, and therefore, opportunities to collect were few (1–3/year). Furthermore, most of the collections were conducted and processed at the housing institutions, and therefore, access to some equipment was not always feasible (e.g., fluorescent microscope for viability assessment). The authors made their best efforts to acknowledge these limitations throughout this manuscript and believe the data provide a groundwork that will guide future studies of polar bear semen collection and cryopreservation.

## 5. Conclusions

In conclusion, UC is an efficacious sperm collection method for polar bears, particularly if conducted during the peak northern hemisphere breeding season of ~March to May. SR is a feasible method of postmortem sperm collection in this species, particularly if the death of the bear occurs during the breeding season. Cryopreservation with egg yolk-based EQ appears to be an effective method for supporting an adequate level of cryosurvival, though further efforts should be made to increase post-thaw sample quality. Additional trials testing the effectiveness of OP are needed before any definitive conclusions can be made regarding its usefulness as an extender for polar bear sperm. Overall, these data provide a solid foundation for future studies investigating polar bear sperm collection and cryopreservation and support efforts to preserve valuable genetic material from this rare species.

## Figures and Tables

**Figure 1 animals-12-00430-f001:**
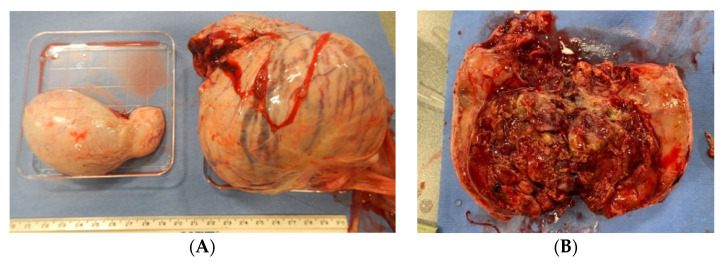
Gross polar bear testicular anatomy: (**A**) an averagely sized testis (70 mm × 45 mm; left) displayed next to a testis with a tumor (125 mm × 100 mm; right); (**B**) a cross section of the testis with tumor.

**Table 1 animals-12-00430-t001:** Demographics, sperm collection method (*n* = 38 collections), and outcome per individual polar bear (*n* = 17) collected via electroejaculation (EEJ), urethral catheterization (UC), or sperm rescue (SR). Proven is defined as successfully bred with females resulting in cub production.

Male	Age	Collection Method	Reproductive		Presence of Sperm	Cryopreserved?
			Season	Status		
1	24	EEJ	Breeding	proven	Yes	Yes
	28	UC	Breeding		Yes	No
	32	SR *	Non		No	---
2	10	EEJ	Breeding	unproven	No	---
	11	EEJ	Non		No	---
	11	UC	Non		No	---
	11	SR	Non		No	---
3	25	SR	Non	proven	Yes	Yes
4	31	SR *	Breeding	unproven	Yes	Yes
5	24	UC	Non	proven	Yes	Yes
	27	UC	Non		Yes	No **
	29	SR *	Breeding		Yes	No
6	22	EEJ	Breeding	unproven	No	---
	22	UC	Breeding		Yes	No
	26	SR	Breeding		Yes	Yes
7	6	EEJ	Breeding	unproven	No	---
	6	UC	Breeding		Yes	Yes
	9	UC	Breeding		Yes	Yes
	12	UC	Breeding	proven	Yes	Yes
8	3	EEJ	Breeding	unproven	No	---
	8	UC	Breeding		Yes	No
	14	UC	Breeding		Yes	No
9	8	UC	Breeding	unproven	Yes	No
10	15	UC	Breeding	unproven	Yes	Yes
	15	UC	Breeding		No	---
	17	UC	Breeding		Yes	No **
	19	UC	Non		No	---
	21	UC	Non	proven	No	---
11	18	UC	Breeding	unproven	Yes	No **
12	10	UC	Breeding	unproven	Yes	Yes
	12	UC	Breeding		Yes	Yes
13	11	UC	Breeding	unproven	Yes	Yes
	12	UC	Breeding		Yes	Yes
14	4	UC	Non	unproven	No	---
	7	UC	Breeding		Yes	No
15	6	UC	Breeding	unproven	No	---
16	26	UC	Breeding	proven	No	---
17	24	SR *	Non	proven	No	---

--- No sperm available for cryopreservation. * Tumor found during SR. ** Sample was urine contaminated.

**Table 2 animals-12-00430-t002:** Polar bear sperm characteristics of samples collected via electroejaculation (EEJ), urethral characterization (UC), or postmortem sperm rescue (SR). Data do not include values from UC samples that were contaminated with urine (*n* = 3).

	EEJ (*n* = 1)	UC (*n* = 15)	SR (*n* = 4)
Sperm motility (%)	50	64.3 ± 7.4	56.7 ± 8.8
Progressive status (0–5 scale)	3	2.8 ± 0.4	2.3 ± 0.3
Sperm viability (%)	77	80.9 ± 3.8 *	80.5 ± 0.5
Morphologically normal (%)	34	47.1 ± 6.6 **	25.0 ± 0.3
Sperm concentration (×10^6^/mL)	202	457.1 ± 237.1	
Volume (µL)	<50	240.2 ± 76.6	<100 ***

* Viability data were only collected for seven procedures. ** Morphology assessments were only conducted for 11 procedures. *** Samples were collected via flushing with extender, and therefore, volume is estimated.

**Table 3 animals-12-00430-t003:** Polar bear sperm characteristics (motility, progression, viability) immediately following collection (“initial”) via electroejaculation (EEJ), urethral catheterization (UC), or sperm rescue (SR) following cryopreservation (“post-thaw”) with equine egg yolk-based extender (EQ) or OptiXcell (OP). Data are presented as mean ± SEM.

Sample Size (n)	Collection Technique	Extender	Initial			Post-Thaw		
			Motility	Progression	Viability *	Motility	Progression	Viability
1	EEJ	EQ + 4% glycerol	50	3	77	1	1	44
9	UC *	EQ + 4% glycerol	74.4 ± 3.6	3.2 ± 0.3	81.3 ± 3.6	35.7 ± 5.2	1.7 ± 0.2	55.5 ± 5.3
1		OP	80	4	86	40	2	57
2	SR	EQ + 4% glycerol	55.0 ± 15.0	2.5 ± 0.5	81 ^	15.0 ± 5.0	1.5 ± 0.5	34.0 ± 21.0
1		OP	60	2	80	0	0	16

* One of the UC collections was split and used to test both EQ and OP. ^ Viability was only collected for one of the two samples.

## Data Availability

The data presented in this study are available on request from the corresponding author. The data are not publicly available due to ethical concerns surrounding the release of medical information without permission from veterinarians and owning institutions.

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
