# Peer review of "Case Studies in Polar Bear (Ursus maritimus) Sperm Collection and Cryopreservation Techniques"

_animals, 2022, doi:10.3390/ani12040430_

Round 1

Reviewer 1 Report

This manuscript evaluated effects of different semen collection methods and cryopreservation extenders on sperm characteristics of polar bear during breeding and non-breeding season. Although there are some limitation for the number of samples and experiment design due to the difficulty for getting samples of rare species, this manuscript provides a foundation for future studies investigating polar bear sperm collection and cryopreservation. Furthermore, the authors pointed out the limitation of this study in the discussion. Therefore, the revision is suggested to be acceptable for publication. 

Author Response

Thank you very much for your review. We greatly appreciate the time and effort you've put into helping us prepare this report for publication.

Reviewer 2 Report

All suggestions and comments have been addressed in this revised MS.

Author Response

(The authors gave the same response as above.)

Reviewer 3 Report

The only note on the manuscript version is "A Fisher's exact test".
What is "A Fisher's exact test"?
The post-hoc test used should be reported.

Author Response

Thank you very much for your review. We appreciate the time and effort you've put into helping us prepare this report for publication.

A Fishers exact test is a non-parametric test, that is analogous to a chi-square test, which allows one to determine if frequencies are similar but can be used when sample sizes are small. Post-hoc tests are not needed because only two groups are being compared at a given time. UC success rate was compared to EEJ success rate and breeding season rate was compared to non-breeding season rate.

This manuscript is a resubmission of an earlier submission. The following is a list of the peer review reports and author responses from that submission.

Round 1

Reviewer 1 Report

Over the last decade, the author has conducted 31 opportunistic sperm collection attempts on 14 male polar bears, to define sperm characteristics, assess if the likelihood of a successful collection was impacted by collection method or season, and evaluate semen extenders for use in polar bear sperm cryopreservation. The research work is time-consuming and difficult, and those data from the research provided a foundation for future studies on polar bear sperm collection and cryopreservation. However, there are some major concerns which affect its publication.

  1. The manuscript studies the influence of factors such as different collection methods, different seasons and different dilutions on semen quality. However, there is no statistical significance for most results, since the sample size is less than 3.
  2. Since the EEJ group has only one value, it is meaningless to compare with the CU group.
  3. There is no duplication in the EEJ group, what is the comparison method in Table 1?
  4. The content of Table 2 does not seem to be complete, and the length and width of testis are not reflected in the table.
  5. Although the manuscript examines the effects of different collection methods, dilutions, and freezing methods on the quality of semen, there is a lack of sufficient data for comparison, and it is doubtful whether the experimental design is scientific.
  6. Please describe the results of the experiment in points to make the research content clearer.
  7. Please avoid using uncertain words such as “anecdotal reports” in line 310.
  8. The content of discussion in the manuscript need to be concise.
  9. In the conclusion, please describe the main findings and significance briefly.

Reviewer 2 Report

The MS on the evaluation of polar bear sperm collection and cryopreservation reports on the comparison between two collections methods:  electroejaculation (EJ) and urethral catheterization (UC). In addition authors tested 5 extenders, 3 cryoprotectants at different concentrations in 10 semen samples. Yet most of those variants tested were individual cases far from a paired comparison of variants. Thus the data on cryopreservation represents a collection of single case reports from which it is hard to draw any conclusions. Only exception is a set of 9 samples collected by catheterization and cryopreserved with equine extender using 4% glycerol. Even in this set of 9 sample it is not clear how many of those are from the same male and represent a repeated sample of the same animal.

Main technical flaw of the study is that authors used an electroejaculation probe far too small for the size of a polar bear. The method described for electroejaculation is insufficient to collect sperm from a polar bear. The probe described in the methods is the same size as is used in humans. In an adult polar bear (+500 kg), it is not surprising that a small probe of 3.3cm did not produce any ejaculate even during breeding season. As the MS’s statistical analysis, results and conclusions is built around the comparison between EJ and UC most of the MS becomes invalid.

Yet, as data on semen collection and cryopreservation in this species is scarce, I suggest to present the data as a technical note just reporting on the use of urethral catheterization as alternative method to collect semen using EQ + 4% glycerol as a first promising start to sperm cryopreservation in polar bears. The MS should be edited accordingly.

line 113-116/201-203/211-216/266-272 : probe is to small which in turn is responsible for the failure to produce an ejaculate. In all other references, specifically giant panda, EJ produces good quality ejaculates. Explanation of EJ failure in polar bears as provided in the discussion falls short of the discussion of own possible mistakes in the choice of collection technique. Delete EJ data from MS.

line 180: Here two different extenders were mixed. This is scientifically and technically not sound. In addition post thaw results for those collections were at 0-1% motility. This seems a technical problem Thus those results cannot be interpreted at all. Delete.

Line 162-169 / Table 2: Two samples frozen with URF with two different extenders (n=1) showing 0% post thaw motility. These are preliminary results from individual case reports. Conclusions on the use of URF method can not be drawn. Delete from MS.

Line 278: This paper discusses use of EJ without anaesthesia in domestic cattle bulls. This is an inappropriate reference here. Delete. Bears are under general anaesthesia during semen collection so that endured pain during collection is not a reason for alternative collection methods. Duration of anaesthesia when a bear is catheterized three times for semen collection with pauses of 5 min in between adds up to the same time that is needed for an electroejaculation. Authors need to specify the advantages of UC compared to EJ in the case of the polar bear when the mandatory anaesthesia is not shortened.

Table 1: edit: include individual results with means in the bottom line.

Throughout the MS reference no. 14 is cited. This self-cross reference that is not yet accepted or published in a peer-reviewed journal. Data from this self-reference is cited multiple times in the introduction and discussion I suggest to either cite it as unpublished data throughout the MS.

Line 356: questions arises why authors have not chosen to combine results from post mortem and in situ sperm cryopreservation in polar bears. In view of the scarce data summarized here a larger data set might have drawn a more uniform, solid, first picture of polar bear sperm cryopreservation.

Reviewer 3 Report

This article describes the authors' experience in sperm polar bear collection and cryopreservation during the last decade. Despite several limitations (which are recognized and discussed in the manuscript), the study design is clear and it is well-presented. The study is very well-written in English. I think the topic has been poorly investigated so far and this study can be surely considered of interest. Hence, I recommend the publication after minor revision.

Line 62: introduce abbreviations for EEJ and UC techniques, and be consistent along the entire manuscript.

In methods, what AVG means? If authors refer to average, better not to write it capital.

In the results, it is stated: “The two successful collections conducted in the non-breeding season were collected from the same individual who had a testicular tumor [14]. When data from two males (3 collections total) confirmed at necropsy to have testicular tumors [14] were removed from the analysis, the impact of season on likelihood of success increased (P = 0.001).” I do not understand why authors excluded this data from the analysis. The presence of tumor might be responsible of worsening semen quality. However, collections were successful. I encourage the authors to explain the reason (if any) of this exclusion, otherwise data should be included in the final analysis.

Reviewer 4 Report

The topic covered in the article is the most modern and interesting. The availability of semen of species considered endangered and their conservation in view of the storage of their genetic material is one of the major problems of today's biology.

The presented manuscript shows that these are preliminary studies, especially in the selection of diluents and methods of semen freezing, although they cover quite a long period of time (10 years). It seems necessary to add the phrase "A pilot study" in the title.

The failures of EEC seem to completely cancel out this method (n = 1), so it seems inadvisable to compare this method with UC. This is especially true for semen parameters after cryopreservation.

Doubts are also raised when pointing to significant differences between the semen collection season - it is well known that semen obtained outside the reproductive season is characterized by worse quality parameters. The Authors agree with this fact, pointing to differences in the size of the testes of individuals present in the experiment (although the differences were not statistically significant).

The methods used to assess sperm motility are also not very reliable. Apart from the subjective assessment of the overall semen motility, the assessment of the progressive movement of sperm raises doubts. The method indicated by the authors has no literature reference, although it is commonly used by them in other studies. The wide availability of CASA systems would rather require their use in the presented studies, additionally enriching the knowledge with kinematic parameters of sperm movement.

The use of the eosin / nigrosine test to assess the viability of sperm and the poor cell morphology test also seems to be not very accurate with today's microscopic techniques, especially since the Authors use fluorescence techniques in other publications (10.1016/j.theriogenology.2018.07.042).

The use of multiple sperm cryopreservation procedures (extenders) seems superfluous, especially with this number n. The results nevertheless indicated the classical procedure (EQ + 4% glycerol) as the best that could be predicted.

Reference number 14 raises a big problem - it is a mistake to refer to unpublished data. Perhaps the combination of both manuscripts would provide a broader view of the results obtained by the Authors.

Considering the above, the article could only be published as "Short communication".